materials science

poly(butylene succinate), copolymers, properties, degradability, vegetable growth

**Authors for correspondence:**
Min Zhang
e-mail: yanjiushi206@163.com
Zongcheng Miao
e-mail: miaozongcheng@xijing.edu.cn

This article has been edited by the Royal Society of Chemistry, including the commissioning, peer review process and editorial aspects up to the point of acceptance.

# Poly(butylene succinate-co-salicylic acid) copolymers and their effect on promoting plant growth

Lei Wang[1], Min Zhang[1,2], Tom Lawson[3], Aqsa Kanwal[2] and Zongcheng Miao[4]

[1]Shaanxi Key Laboratory of Chemical Additives for Industry, and [2]College of Environmental Science and Engineering, Shaanxi University of Science and Technology, Xi'an 710021, People's Republic of China
[3]ARC Centre of Excellence for Nanoscale Biophotonics, Macquarie University, Sydney, New South Wales 2109, Australia
[4]Key Laboratory of Organic Polymer Photoelectric Materials, Xijing University, Xi'an 710123, People's Republic of China

MZ, 0000-0001-9301-6615; AK, 0000-0001-8040-2453

Biodegradable random copolymers were successfully synthesized by melt polycondensation of poly(butylene succinate) (PBS) and salicylic acid (SA). The obtained copolymers were characterized by proton nuclear magnetic resonance spectroscopy. The effect of different SA contents on the properties of copolymers was investigated by universal testing machine, thermogravimetric analyser, differential scanning calorimetry and X-ray diffraction analysis. The results showed that the copolymers with 0.5% SA contents exhibited excellent elastic modulus (1413.0 MPa) and tensile strength (192.8 MPa), and similar thermal decomposition temperature ($\approx$320°C) compared with pure PBS. By molecular docking simulations, it was proved that the degradability of copolymers was more effective than that of pure PBS with a binding energy of −5.77 kcal mol$^{-1}$. PBS copolymers with a small amount of SA were not only biodegradable but could stimulate the growth of green vegetables. So biodegradable copolymers can be used over a wide range as they are environmentally friendly.

## 1. Introduction

For decades, traditional polymeric materials have been widely used in many fields. Among them, packaging has a considerable market share. As a result, millions of tons of plastic packaging are landfilled every year [1] and cause environmental pollution. This led to the research that focused

on developing biodegradable materials [2,3]. Other environmental friendly substances are now gradually replacing many harmful chemical compounds, like petroleum hydrocarbons [4–7]. Poly(butylene succinate) (PBS) is a biodegradable aliphatic polymer with a relatively high melting temperature, good toughness and processing ability [8–12]. Hence it is widely used in agriculture, industry and other fields. But as these PBS properties are insufficient for its wider range of applications, there is a need to modify PBS to obtain desired properties, such as satisfactory mechanical properties, good biodegradability and processing ability [13–15]. Many previous studies have reported preparation of PBS-based copolymers and their blends. But it is also challenging to recycle these products after being used. The accumulation of PBS residues in soil have neither harmful nor beneficial significant impacts on soil and plants [16,17].

Salicylic acid (SA) is an inexpensive, non-toxic and biodegradable phenolic compound [18], which is frequently used as an anti-inflammatory, antioxidant and antibacterial in medicine [19,20]. SA has been incorporated into medical polymer materials, such as polylactic acid (PLA), poly(anhydride esters) and poly(ε-caprolactone) (PCL), to improve their degradability and compatibility with biological specimens [21,22]. SA containing a phenyl group can change the properties of copolymers including crystallinity, degradation rates and biocompatibility thus increasing their wide range application [23]. In agriculture, SA acts as an endogenous signal molecule in plants' growth [24,25]. It induces crop damage resistance [26–30]. This suggests that SA is beneficial to induce plant growth.

At present, there are few studies on the application of SA-based copolymers in agriculture, but polymers, especially PBS as agricultural mulch films, have been widely studied. It has been reported that only a small amount of SA is sufficient to induce plant growth in agriculture [31] and the released products were found to be non-toxic [19]. SA-based copolymers can release SA continuously during degradation in soil, hence increasing plant growth. After the polymer material gets discarded, it is very important to find a way to minimize soil pollution and make it beneficial to crop growth.

In this study, we focused on preparing copolymers by incorporating SA in PBS polymer that could increase plant growth without affecting comprehensive performance of PBS polymer. Since a small amount of SA can promote the growth of plants, this study mainly discussed the influence of low SA content on copolymers and compared the influence of medium and high SA content on copolymers. The enzymatic degradation of copolymers was investigated. Molecular docking simulation was used to study the molecular chain structures during degradability of copolymers. Capacity of copolymers to improve plant growth was also investigated.

# 2. Materials and methods

## 2.1. Materials

Succinic acid (greater than 99.5 wt%) was purchased from the Sinopharm Chemical Reagents Company. 1,4-butanediol (greater than 99.5 wt%) was purchased from the Tianjin Fuchen Chemical Reagents Company. SA was obtained from the Traditional Chinese Medicine Chemical Reagents Company. Tetrabutyl titanate (Ti(OBu)$_4$) (greater than 99.0 wt%) and chloroform (CHCl$_3$) was supplied by the Tianjin Oumi Chemical Reagents Company. Chemicals were used as received. *Candida antarctica lipase* B (CALB) was purchased from the Novozymes (China) Investment Company.

## 2.2. Synthesis of copolymers

PBS-based copolymers were prepared by the melt polycondensation method as reported earlier [32], including two steps of esterification and polycondensation. Succinic acid, 1,4-butanediol and SA as raw material were added into the three-necked flask at 1 : 1.1 (acid to alcohol), then 0.5 mol% of Ti(OBu)$_4$ was added as the catalyst for the reaction. The reactor was filled with nitrogen to protect the reaction from oxidation. The reactants were heated to melt in the oil bath at 190°C, until the volume of dehydrate and the theoretical amount of H$_2$O became equal. The phenyl functional groups from SA were added to the polymer chain during the dehydration reaction. Then the polycondensation process was carried out controlling the pressure at below 5.5 kPa (vacuum degree). It was continually heated until 220°C and reacted for approximately 3 h to obtain the desired products. Products were dissolved in CHCl$_3$, and purified several times with ethanol, and then vacuum-dried for 24 h.

## 2.3. Characterizations

Proton nuclear magnetic resonance spectroscopy ($^1$H NMR) was obtained with an Avance 400 spectrometer (Bruker, Germany) at 400 MHz. CDCl$_3$ was applied as the solvent. Chemical shifts ($\delta$) were given in ppm with respect to tetramethylsilane (TMS).

Molecular weight data of copolymers was obtained by gel permeation chromatography (GPC) using an EC2006 (Dalian Yilite Analytical Instruments, China) equipped with Shodex RI-201H refractive index detector. Chloroform was used as the mobile phase at a 1.0 ml min$^{-1}$ flow rate and the sample solution was 1.0 w/v%. The column temperature was maintained at 40°C in all analyses. The number-average ($M_n$) and weight-average molecular weights ($M_w$) calibration curves were obtained by means of polystyrene standards.

Mechanical properties of copolymers were measured by uniaxial tensile test on a XWWW-20 (Chengde Jinjian, China) universal testing machine according to GB/T 1040-96 standard [33]. The dumbbell-shaped samples (width 6 mm, useful length 80 mm) were stretched with a crosshead speed of 2 mm min$^{-1}$ and a load cell of 2000 N. All samples were replicated thrice and the average value was computed.

Thermal gravimetric analysis (TGA) was done with a Q500 system (TA, USA) to evaluate the thermal stability of the copolymers. The analysis was carried out in a nitrogen atmosphere. About 5 mg of the sample was heated from 25°C to 500°C at a heating rate of 10°C min$^{-1}$. Derivative thermogravimetry (DTG) curves were also obtained.

Differential scanning calorimetry (DSC) measurements were performed using a Q2000 system (TA, USA) to obtain thermal properties of the copolymers. About 3 mg of sample was heated from 25°C to 200°C at a rate of 30°C min$^{-1}$ in order to eliminate any previous thermal history. The temperature was then reduced to −50°C at a rate of 10°C min$^{-1}$. The temperature was then increased again to 200°C at a rate of 10°C min$^{-1}$. The crystallization temperature ($T_c$) of polymers was determined during the cooling scan and its glass transition temperature ($T_g$), melting temperature ($T_m$) and melting enthalpy ($\Delta H_m$) were measured during the second heating scan.

Polarized optical microscope (POM) was used with a DK-POL (Chongqing Optec, China) in order to observe the crystalline morphologies of the copolymers. Samples were prepared by dipping polymers in chloroform solution on the glass slide and the sample film was obtained after evaporation of the solvent. The samples were heated until melt and then temperature was maintained at 80°C for 3 h.

Wide-angle X-ray diffraction (XRD) analysis was performed by Rigaku D/Max-3c Japan apparatus with Cu K$\alpha$ radiation. The scanning range was from 5° to 45° at a rate of 6° min$^{-1}$ with a step of 0.02° ($\lambda = 0.154$ nm).

Degradation analysis of the samples was carried out in phosphate-buffered saline (pH 7.20 ± 0.01, 0.1 mg ml$^{-1}$) at 45°C. The polymeric films were prepared by melt pressing at 120°C, and then cut into rectangle-shaped pieces of about 20 mm length, 10 mm width and 0.4 mm thickness. These films were immersed in a separate tube in triplicate containing 12 ml of phosphate-buffered saline with CALB (4 mg ml$^{-1}$), and incubated at a constant temperature oscillator at 45°C. The samples were removed at regular intervals, washed with deionized water, and then dried in the vacuum at 50°C to attain a constant weight for analysis. The percentage weight loss was calculated as

$$\%\text{weight loss} = \frac{W_o - W_t}{W_o} \times 100\%, \tag{2.1}$$

where $W_o$ is the weight of the sample before degradation and $W_t$ is the constant weight of the sample after degradation at different times intervals [23].

The optimal conformation in the combination between CALB and substrates was analysed by molecular docking. CALB was set as receptor, and BSB, SaBS, SaSB, SaSSa and BSaS as the substrates to implement docking. BSB, SaBS, SaSB, SaSSa and BSaS represent different molecular fragments in the polymer chain, respectively, where B is 1, 4-butanediol, S is succinic acid, and Sa is salicylic acid (figure 1). Before the docking analysis, all the substrates were minimized by ChemOffice v8.0, MM2 force field. The grid energy calculation file was carried out by AutoDock 4.2 tools [34]. The docking box of $60 \times 60 \times 60$ Å was used in each calculation. For each ligand–receptor combination, binding results were obtained by analysing the 200 docking poses. According to the scoring function and the optimal dimensional scale, the lowest binding energy was output to understand the conformation of interaction.

Vegetable was cultivated with a SPX-GB-250 system from Shanghai Botai Instruments. The experimental soil was collected from a local garden. P(BS-co-SA) was added in soil with two levels (0.1 g P(BS-co-SA)/500 g soil and 0.6 g P(BS-co-SA)/500 g soil) and replicated thrice. Likewise the

**Figure 1.** Schematics of (*a*) BSB, (*b*) SaBS, (*c*) SaSB, (*d*) SaSSa and (*e*) BSaS.

**Scheme 1.** Synthesis route of the copolymers P(BS-*co*-SA).

experimental soil of PBS was prepared by the same method. Then the vegetable seedlings with two leaves were transplanted to the experimental soil and cultivated in a light incubator for 60 days at 25°C. The light incubator was programmed to follow a 12 h : 12 h light : dark cycle. Vegetable growth was observed at different times.

# 3. Results and discussion

## 3.1. Synthesis and characterization of copolymers

A series of P(BS-*co*-SA) copolymers was synthesized through melt polycondensation and the feed molar ratio of SA in the copolymers was 0%, 0.5%, 5%, and 10%, respectively. This synthesis procedure is illustrated in scheme 1. The molecular weight and distribution of every P(BS-*co*-SA) copolymer was tested (table 1 and figure 2*b*).

Figure 2*a* shows $^1$H NMR spectra of the copolymers P(BS-*co*-SA). The peak of PBS is in the $\delta$ 1.72 (a), $\delta$ 2.65 (b) and $\delta$ 4.14 (c) spectrum [35]. The double peak in $\delta$ 6.99 (d) and $\delta$ 7.85 (d′), and the triple peak in $\delta$ 6.91 (e′) and $\delta$ 7.47 (e) are illustrated, that show the skeletal units of SA in a benzene ring. These results supported the synthesis of P(BS-*co*-SA).

## 3.2. Mechanical properties

The study of the mechanical properties is very important for P(BS-*co*-SA) application. The results of mechanical properties are detailed in table 2. An apparent increase in elastic modulus of the P(BS-*co*-SA) films from 779.6 to 1400 MPa was observed when SA was added, but there was significant reduction in elongation at break from 13.2% to 2.7%. The tensile strength of the P(BS-*co*-0.5%SA), 192.8 MPa, was more than that of pure PBS (180.5 MPa). P(BS-*co*-SA) films became too fragile to be tested when SA content was 10 mol%. These results displayed a high compatibility between the PBS and the SA when it was added in small amounts. This could be due to a deterioration in the mobility of the chains as the SA content increased, which led to a significant reduction in the flexibility of PBS

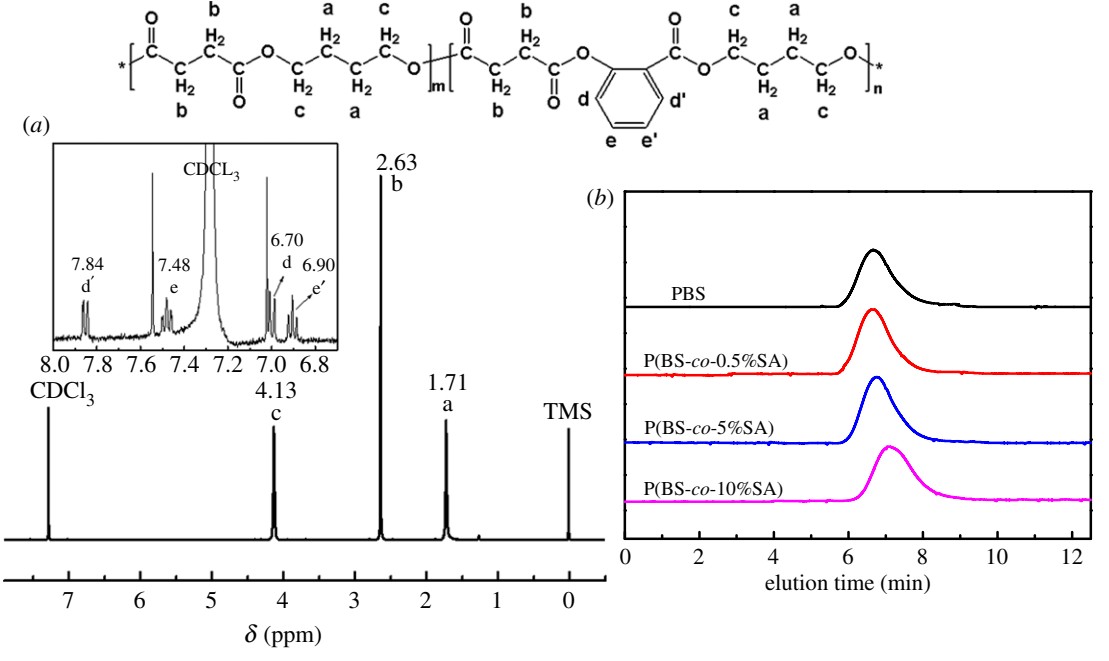

**Figure 2.** (a) $^1$H NMR spectra and (b) GPC diagrams of the copolymers P(BS-co-SA).

**Table 1.** Composition and relative parameters[a] of the copolymers P(BS-co-SA).

| no | polymer | $M_n$[b] kDa | $M_w$[c] kDa | PDI[d] |
|----|---------|------------|------------|--------|
| 1 | PBS | 38.8 | 80.2 | 2.07 |
| 2 | P(BS-co-0.5%SA) | 39.9 | 86.0 | 2.15 |
| 3 | P(BS-co-5%SA) | 31.1 | 67.3 | 2.16 |
| 4 | P(BS-co-10%SA) | 13.5 | 35.1 | 2.61 |

[a]Obtained by GPC calibrated by polystyrene standards.
[b]Number average molecular weight.
[c]Weight average molecular weight.
[d]Polydispersity index.

[36]. The fragility of copolymers was due to the rigidity of its benzene ring, which increased the copolymer's steric hindrance as SA content increased [37].

## 3.3. Thermal analysis

DSC and TGA were used to evaluate the thermal properties of copolymers. The thermal properties are listed in table 3. Figure 3a,b exhibits the DSC curves measured with pure PBS and P(BS-co-SA). It is observed that pure PBS displays a crystallization peak ($T_c$) at 79.4°C, an endothermic peak ($T_m$) at 116.4°C and a glass transition temperature ($T_g$) at −28.5°C. P(BS-co-0.5%SA) was observed to have a high $T_c$ of 83.2°C compared with pure PBS. The $T_g$ (−28.2°C) and $T_m$ (116.7°C) of P(BS-co-0.5%SA) were found to be similar to pure PBS, but $\Delta T$ (33.5°C) was lower than that of pure PBS. These results suggested that the thermal stability of PBS was significantly enhanced when 0.5% SA was added. However, $T_m$ (115.2°C), $T_c$ (77.0°C) and $T_g$ (−29.0°C) of P(BS-co-10%SA) were less compared with PBS, which may be attributed to its lower molar mass [39]. The structural rigidity of the polymer chain increased with the insertion of phenyl groups of SA. But the addition of excess benzene groups disturbed the consistency of the original PBS chain and made the PBS molecular arrangement adjustments. The segment motion activity of PBS chain decreased and led to the result that thermal behaviour of the copolymers became weaker.

The thermal stability of the copolymers is detailed in figure 3c, including a set of DTG curves (figure 3d). It showed the quickest decomposition temperature of P(BS-co-SA) copolymers was at

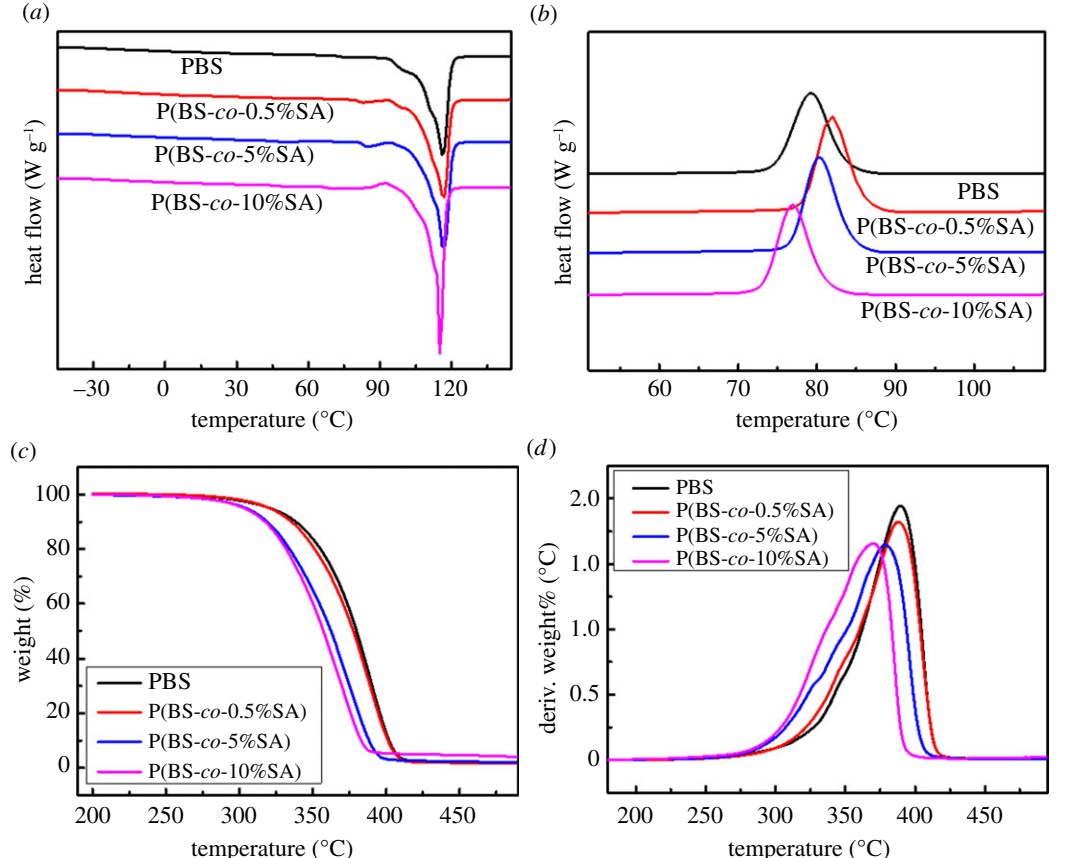

**Figure 3.** DSC and TGA curves of P(BS-*co*-SA): (*a*) second heating scan, (*b*) cooling scan, (*c*) TGA curves and (*d*) DTG curves.

**Table 2.** Mechanical properties of the P(BS-*co*-SA) films. ('—' indicates film that was too fragile to be tested.)

| polymers | elastic modulus (MPa) | tensile strength (MPa) | elongation at break (%) |
|---|---|---|---|
| PBS | 779.6 ± 4.65 | 180.5 ± 2.87 | 13.2 ± 1.54 |
| P(BS-*co*-0.5%SA) | 1413.0 ± 5.04 | 192.8 ± 2.43 | 11.3 ± 1.98 |
| P(BS-*co*-5%SA) | 1434.7 ± 4.18 | 87.9 ± 1.18 | 2.7 ± 2.12 |
| P(BS-*co*-10%SA) | — | — | — |

**Table 3.** Thermal performance of various P(BS-*co*-SA) determined by TGA and DSC.

| polymer | $T_g$ (°C) | $T_m$ (°C) | $\Delta H_m^a$ (J g$^{-1}$) | $\chi_c^b$ (%) | $T_c$ (°C) | $\Delta T$ (°C) | $T_{5\%}^c$ (°C) | $T_{50\%}^d$ (°C) |
|---|---|---|---|---|---|---|---|---|
| PBS | −28.5 | 116.4 | 76.6 | 69.3 | 79.4 | 37.0 | 321.2 | 379.0 |
| P(BS-*co*-0.5%SA) | −28.2 | 116.7 | 72.9 | 66.0 | 83.2 | 33.5 | 320.6 | 377.4 |
| P(BS-*co*-5%SA) | −26.9 | 116.5 | 75.0 | 67.9 | 80.3 | 36.2 | 304.3 | 362.8 |
| P(BS-*co*-10%SA) | −29.0 | 115.2 | 75.7 | 68.5 | 77.0 | 38.2 | 303.2 | 357.3 |

[a]Determined by the second heating scan at 10 min$^{-1}$.
[b]The crystallinity ($\chi_c$) of copolymers were calculated by $\chi_c = \Delta H_m / \Delta H_m^o \times 100\%$, and $\Delta H_m^o$ is the melting enthalpy of 100% crystalline PBS, which is assumed to be 110.5 J g$^{-1}$ [38].
[c]Decomposition temperature of copolymers at weight loss of 5%.
[d]Decomposition temperature of copolymers at weight loss of 50%.

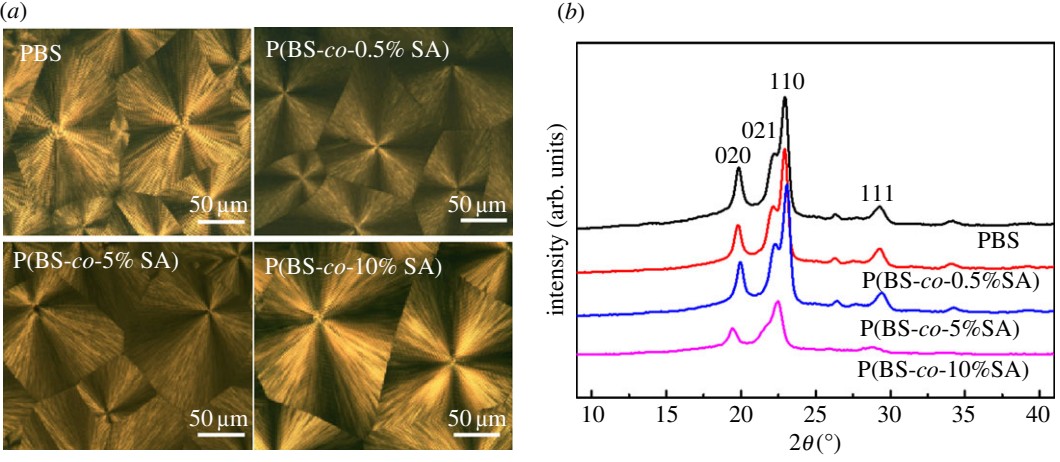

**Figure 4.** (a) Optical micrographs of the spherulitic morphology of P(BS-co-SA) and (b) XRD diffractograms of P(BS-co-SA).

300°C. Its disintegration temperature range was broader. It was observed that P(BS-co-0.5%SA) have similar TGA traces with pure PBS. As the SA content increased, the initial decomposition temperature of the copolymers decreased from 321.2°C to 303.2°C. The exceptional thermal stability of P(BS-co-SA) copolymers remained nearly unaffected with changes to the amount of SA added. Similarly, decomposition temperature remained above 300°C.

## 3.4. Crystalline behaviour

The crystalline morphology of P(BS-co-SA) samples was investigated by POM during isothermal crystallization (figure 4a). After isothermal crystallization for 1 h at 80°C, the copolymers showed a similar ability to nucleate and grow. Both crystals of PBS and P(BS-co-SA) were spheroidal, and spherulites covered the bulk [40]. It is clear that the spherulite size of the P(BS-co-SA) is similar to all samples except P(BS-co-10%SA). It showed that SA could induce nucleation sites for P(BS-co-SA) crystallization and change the crystallization behaviour of P(BS-co-SA) [41], probably due to the rigidity of benzene ring, which decreased P(BS-co-SA) chain mobility [42]. In other words, the relatively large size of the P(BS-co-10%SA) spherulite was because of few nucleation points. These results were similar with the low $T_g$ in DSC measurement, displaying the difficulty of nucleation with the addition of SA.

The XRD diffractograms of the P(BS-co-SA) copolymers are shown in figure 4b. The graphs of the copolymer have similar shape. A slow change in diffraction peaks can be observed. Pure PBS has diffraction peaks at 020 (19.8°), 021 (22.2°) and 110 (22.9°). A smaller diffraction peak can be seen at 111 (29.2°) [43]. The diffraction peaks of P(BS-co-SA) match those of PBS. The intensities of P(BS-co-SA) diffraction peaks are the same as that of pure PBS, meanwhile, the locations of diffraction peaks are almost the same. It suggested that their crystal structures are similar. When a copolymer was made from 10 mol% SA its diffraction peak became smaller. Its position offset was towards the small angle. It suggested significant difference in crystallinity. A large amount of SA included in the main chain of the PBS disturbed its neat double helix arrangement and its ordered structure become more chaotic.

## 3.5. Enzymatic degradation and molecular docking

When SA is released in soil, it is mainly affected by two factors: hydrolytic and microbial degradation of polymers. Hence the degradation performance of P(BS-co-SA) films was explored by enzymatic degradation with hydrolysis. Moreover, hydrolytic degradation is more rapid in basic media than acidic and alkaline media. At pH 7, salicylic acid was rapidly released [21].

The degradability of P(BS-co-SA) films in basic media was studied as shown in figure 5. It was found in enzymatic degradation that % weight loss of P(BS-co-SA) was significantly higher compared with pure PBS. At day 20, % weight loss of P(BS-co-5%SA) was recorded as 3.38%, which is considerably higher than 1.53% of pure PBS. It was because the spiral chain structure of PBS had been destroyed and the crystallinity of PBS was changed in the presence of SA [44]. The chain of P(BS-co-0.5%SA) was considerably more flexible than that of PBS. It is beneficial for CALB to attack the ester bond [45] to

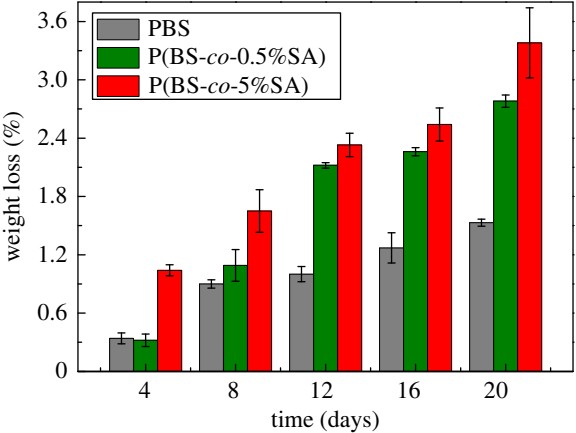

**Figure 5.** The degradation evaluation of P(BS-*co*-SA) in phosphate buffer solution containing CALB.

**Table 4.** Docking results of substrate molecules by AutoDock program. The binding results were obtained by analysing the 200 docking poses performed by AutoDock 4.2.

| enzyme | ligand | $E_{binding}^{a}$ (kcal mol$^{-1}$) | $E_{inter\text{-}mol}^{b}$ (kcal mol$^{-1}$) | $E_{total\text{-}internal}^{c}$ (kcal mol$^{-1}$) | $E_{torsional}^{d}$ (kcal mol$^{-1}$) | $E_{vdW}^{e}$ (kcal mol$^{-1}$) | $E_{elec}^{f}$ (kcal mol$^{-1}$) |
|---|---|---|---|---|---|---|---|
| CALB | BSB | −4.04 | −8.51 | −1.25 | 4.47 | −8.33 | −0.18 |
| | SaBS | −5.00 | −8.88 | −0.58 | 3.88 | −8.87 | −0.01 |
| | SaSB | −5.48 | −9.36 | −1.43 | 3.88 | −9.21 | −0.15 |
| | SaSSa | −5.77 | −9.05 | −2.78 | 3.28 | −8.92 | −0.13 |
| | BSaS | −4.40 | −8.27 | −2.43 | 3.88 | −8.08 | −0.20 |

[a]Binding energy.

[b]Intermolecular energy.

[c]Total internal energy.

[d]Torsional energy.

[e]Van der Waals energies.

[f]Electrostatic energy.

result in degradation of P(BS-*co*-0.5%SA). In addition, it is easy for enzymes to attack molecular chains as SA increased, resulting in efficient degradation by P(BS-*co*-5%SA) [46].

The interactions between CALB lipase and ligands were explained at the molecular level by molecular docking simulations. The binding free energy was calculated and the relationships among all related energies are clearly shown in table 4. Among them the $E_{torsional}$ only depends on the number of torsion bonds and $E_{unbond}$ is equal to $E_{total\text{-}internal}$. In accordance with the molecular docking results, the binding free energy of compounds formed by substrates (SaBS, SaSB, SaSSa and BSaS) with CALB lipase were −5.00, −5.48, −5.77 and −4.40 kcal mol$^{-1}$, respectively, which were lower than that of BSB, −4.04 kcal mol$^{-1}$. The lower the binding energy, the more stable the substrate binding to the enzyme [45]. The binding free energy of SaSSa with CALB lipase was the lowest, so SaSSa was the most stable with CALB lipase. Then the substrate containing SaBS, SaSB and BSaS units was relatively stable with CALB lipase owing to the lower binding energy. It can be concluded from the above results that the presence of the SA unit could significantly reduce binding energy and enhance the degradability of PBS in phosphate buffer solution. Therefore, the enzymatic degradability of P(BS-*co*-SA) was more effective than that of PBS.

## 3.6. Effects on vegetable growth

We investigated the effect of soil-incorporated copolymers' residues on the growth of plants. The effect of different levels of soil-incorporated P(BS-*co*-SA) copolymers on vegetable growth was studied (figure 6). Three levels (0, 0.1 and 0.6 g) of P(BS-*co*-SA) were used during the study. Soil incorporated without

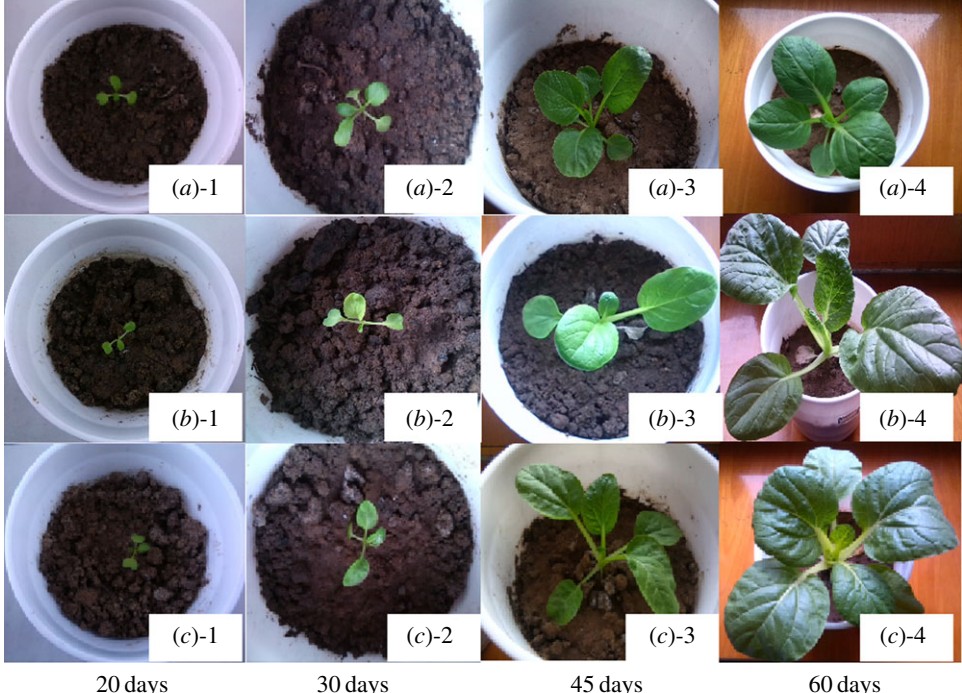

**Figure 6.** Chinese cabbage growth in soil containing (*a*) PBS, (*b*) 0.1 g P(BS-*co*-SA), (*c*) 0.6 g P(BS-*co*-SA) after various time periods.

P(BS-*co*-SA) was used as control soil. It can be seen from figure 6 that no significant change in plant growth was observed during the first 20 days. At day 30, the number of leaves recorded in A-2 was higher than in B-2 and C-2. But with the time, treatments containing P(BS-*co*-SA) showed a higher growth rate compared with the treatments without P(BS-*co*-SA) (control treatment). This was because P(BS-*co*-SA) undergoes little or no degradation in the early days because release of SA into the soil from the polymer takes time [36]. At day 60, it was found that the number of leaves in all treatments was the same but the plants in the soil treated with copolymers exhibited larger and green leaves compared with control. This became more obvious with the increase in copolymer content. After a period of time, the P(BS-*co*-SA) hydrolyses upon exposure to aqueous conditions in soil, releasing SA while leaving the PBS backbone fracture for degradation [47]. The vegetable growth for different SA content showed that the polymer with the highest content of SA had a better relative growth rate. P(BS-*co*-SA) stimulated plant growth and no toxicity was observed. It is well-known that SA can induce vegetable growth. The results reported here agree with this consensus.

## 4. Conclusion

A series of P(BS-*co*-SA) copolymers was synthesized successfully by a two-step melt polycondensation reaction, which has a superior ability to induce plant growth compared with PBS. The effects of low, medium and high SA content on copolymers' performance were investigated. Tensile tests and thermal evaluation showed that copolymers had good mechanical and thermal properties when a small amount of SA was added. Thermal stability up to 300°C was confirmed by TGA. XRD indicated that the copolymers with different SA contents had similar crystal structures. The enzymatic degradation of P(BS-*co*-SA) was faster than that of PBS in phosphate buffer solution. The enzymatic degradation fit well with the molecular docking results, the presence of the SA unit could significantly enhance the degradability of PBS due to the lower free binding energy, indicating the favourability of adding SA. These results suggested that adding a small amount of SA would not affect the comprehensive performance of PBS, however, it could stimulate the growth of green vegetables and can be used as environmentally friendly material.

Data accessibility. All data and research materials supporting the results are in the article.

Authors' contributions. L.W. and M.Z. conceived the study and designed the study. L.W. carried out the laboratory work, participated in data analysis, participated in the design of the study and drafted the manuscript. T.L. and A.K. reviewed and edited the paper. Z.M. participated in data analysis. All authors gave final approval for publication.

Competing interests. We have no competing interests.

Funding. This study was supported by the National Natural Science Foundation of China (grant no. 51673157) and the Natural Science Basic Research Plan in Shaanxi Province of China (grant no. 2017JM5134 and 2018JM5047).

Acknowledgements. The authors would like to thank Xiaoning Ma for her help and guidance in molecular docking.

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
