## [Reviewer comments · Royal Society Open Science]

Review History

RSOS-190504.R0 (Original submission)

Review form: Reviewer 1

Is the manuscript scientifically sound in its present form?

Yes

Are the interpretations and conclusions justified by the results?

Yes

Is the language acceptable?

Yes

Is it clear how to access all supporting data?

Yes

Do you have any ethical concerns with this paper?

No

Have you any concerns about statistical analyses in this paper?

No

Recommendation?

Accept with minor revision (please list in comments)

Comments to the Author(s)

1. The authors should clarify how to remove $Ti(OBu)_4$ when purified P(BS-co-SA) product.
2. The authors should give the detailed information for the BSB, SaBS, SaSB....
3. The authors should give the degradation mechanism of PBS and P(BS-co-SA) for readers understanding.

Review form: Reviewer 2

Is the manuscript scientifically sound in its present form?

Yes

Are the interpretations and conclusions justified by the results?

Yes

Is the language acceptable?

Yes

Is it clear how to access all supporting data?

Yes

Do you have any ethical concerns with this paper?

No

Have you any concerns about statistical analyses in this paper?

No

Recommendation?

Major revision is needed (please make suggestions in comments)

Comments to the Author(s)

The work reported in this manuscript is interesting and well presented. However, it needs improvements before the acceptance. The work requires revision. Some comments are:

1. The title should be improved; it should be more precise and represent to the contents.
2. In the abstract; "proved that the degradability of copolymers was more." add quantitative data
3. The abstract should be re-written to summarize the work; the abstract should state briefly the purpose of the research, the principal results and major conclusions. An abstract is often presented separately from the article, so it must be able to stand alone
4. The Introduction should be focused and goes over the topic
5. Introduction should increase scientific understanding sufficiently, to be of a very high quality paper, what is the gap to cover
6. All abbreviations should be defined
7. Page 5 Section 2.2. Synthesis of copolymers, should be revised with more details and steps are to be clear
8. Scheme 1 is to be revised, add all conditions, T, used gas, media, aqueous / not, time, initiator,

etc

9. Section 2.3. Characterizations. Please add more details about the procedure; check the numbers. Be more specific, what about the control set

10. Please correct the errors and typos

11. Please correct the equations

12. Please improve Fig. 2. with labeling and descriptions; please revise the interpretation and correct

13. Please improve the discussion of molecular docking

14. Effects on Vegetable Growth: please discuss it deeply as it is important section

15. There are many sentence that required to be clarified.

16. Under introduction after "Many environmentally unfriendly chemical compounds like petroleum hydrocarbons" please add please add Ecological Engineering 120, 126-133; IGI ISBN13: 978152252136; Elsevier ISBN-13: 978-0128047033; Journal of Environmental Chemical Engineering 6 (4), 5361-5368; Journal of environmental management 211, 323-333

17. More emphasis on finding and its implication may be mentioned in the conclusion section.

18. Captions of the figures must be with complete information, experimental conditions etc

19. Make sure all analysis was done at least three times with no significant difference

20. Quality of figures is to be improved, please improve the figures

21. Values in the tables should be of uniform significant figures, please recheck

I WOULD LIKE TO SEE THE REVISED MANUSCRIPT.

Review form: Reviewer 3

Is the manuscript scientifically sound in its present form?

Yes

Are the interpretations and conclusions justified by the results?

Yes

Is the language acceptable?

No

Is it clear how to access all supporting data?

Yes

Do you have any ethical concerns with this paper?

No

Have you any concerns about statistical analyses in this paper?

No

Recommendation?

Major revision is needed (please make suggestions in comments)

Comments to the Author(s)

In this manuscript, the authors presented work on designing and synthesizing the copolymer containing salicylic acid (SA) motifs via polycondensation method. It seems that they successfully obtained the copolymer and evaluated their basic properties. The data presented in the paper are of relatively high quality. After evaluating the overall topic of the work, which seems to address an important of the field, however, several issues need to be clarified before accepted in this

journal.

- a) Please show the GPC traces in the manuscript.
- b) Please polish your writing, for example, the sentence "The degradation rate of P(BS-co-SA) was high in enzymatic solution" in the manuscript should be deleted.
- c) The author stated that all samples were replicated thrice, please add error bar in the Tables and Figure 5.
- d) The author mentioned "spiral chain structure of PBS...", please add relative references.
- e) To intuitively grasp the properties difference of different samples, the author should use the experimental data to state the properties change, for example, "The tensile strength of the copolymers was less than pure PBS except for P(BS-co-0.5%SA)"; "Figure 3A and 3B exhibit the DSC curves measured with pure PBS and P(BS-co-SA). P(BS-co-SA) copolymers were observed to have similar T_m and a high T_c compared with pure PBS," there are too many sentences should be stated using the experimental data.
- f) please explain this sentence "but their location is offset in the direction of the large angle. It suggested that their crystal structures are similar. However, crystal behavior is not similar".

Decision letter (RSOS-190504.R0)

30-Apr-2019

Dear Professor Zhang:

Title: Investigation of Biodegradable Poly(butylene succinate-co-salicylic acid) Copolymers with Promoting Plant Growth
Manuscript ID: RSOS-190504

The editor assigned to your manuscript has now received comments from reviewers. We would like you to revise your paper in accordance with the referee and Subject Editor suggestions which can be found below (not including confidential reports to the Editor). Please note this decision does not guarantee eventual acceptance.

Please submit your revised paper before 23-May-2019. Please note that the revision deadline will expire at 00.00am on this date. If we do not hear from you within this time then it will be assumed that the paper has been withdrawn. In exceptional circumstances, extensions may be possible if agreed with the Editorial Office in advance. We do not allow multiple rounds of revision so we urge you to make every effort to fully address all of the comments at this stage. If deemed necessary by the Editors, your manuscript will be sent back to one or more of the original reviewers for assessment. If the original reviewers are not available we may invite new reviewers.

When submitting your revised manuscript, you must respond to the comments made by the referees and upload a file "Response to Referees" in "Section 6 - File Upload". Please use this to

document how you have responded to the comments, and the adjustments you have made. In order to expedite the processing of the revised manuscript, please be as specific as possible in your response.

Please also include the following statements alongside the other end statements. As we cannot publish your manuscript without these end statements included, if you feel that a given heading is not relevant to your paper, please nevertheless include the heading and explicitly state that it is not relevant to your work.

• Ethics statement

Please clarify whether you received ethical approval from a local ethics committee to carry out your study. If so please include details of this, including the name of the committee that gave consent in a Research Ethics section after your main text. Please also clarify whether you received informed consent for the participants to participate in the study and state this in your Research Ethics section.

OR

Please clarify whether you obtained the necessary licences and approvals from your institutional animal ethics committee before conducting your research. Please provide details of these licences and approvals in an Animal Ethics section after your main text.

OR

Please clarify whether you obtained the appropriate permissions and licences to conduct the fieldwork detailed in your study. Please provide details of these in your methods section.

RSC Associate Editor:
Comments to the Author:
(There are no comments.)

RSC Subject Editor:
Comments to the Author:
(There are no comments.)

Reviewers' Comments to Author:

Reviewer: 1

Comments to the Author(s)

1. The authors should clarify how to remove $Ti(OBu)_4$ when purified P(BS-co-SA) product.
2. The authors should give the detailed information for the BSB, SaBS, SaSB....
3. The authors should give the degradation mechanism of PBS and P(BS-co-SA) for readers understanding.

Reviewer: 2

Comments to the Author(s)

The work reported in this manuscript is interesting and well presented. However, it needs improvements before the acceptance. The work requires revision. Some comments are:

1. The title should be improved; it should be more precise and represent to the contents.
 2. In the abstract; "proved that the degradability of copolymers was more." add quantitative data
 3. The abstract should be re-written to summarize the work; the abstract should state briefly the purpose of the research, the principal results and major conclusions. An abstract is often presented separately from the article, so it must be able to stand alone
 4. The Introduction should be focused and goes over the topic
 5. Introduction should increase scientific understanding sufficiently, to be of a very high quality paper, what is the gap to cover
 6. All abbreviations should be defined
 7. Page 5 Section 2.2. Synthesis of copolymers, should be revised with more details and steps are to be clear
 8. Scheme 1 is to be revised, add all conditions, T, used gas, media, aqueous / not, time, initiator, etc
 9. Section 2.3. Characterizations. Please add more details about the procedure; check the numbers . Be more specific, what about the control set
 10. Please correct the errors and typos
 11. Please correct the equations
 12. Please improve Fig. 2. with labeling and descriptions; please revise the interpretation and correct
 13. Please improve the discussion of molecular docking
 14. Effects on Vegetable Growth: please discuss it deeply as it is important section
 15. There are many sentence that required to be clarified.
 16. Under introduction after "Many environmentally unfriendly chemical compounds like petroleum hydrocarbons" please add please add Ecological Engineering 120, 126-133; IGI ISBN13: 978152252136; Elsevier ISBN-13: 978-0128047033; Journal of Environmental Chemical Engineering 6 (4), 5361-5368; Journal of environmental management 211, 323-333
 17. More emphasis on finding and its implication may be mentioned in the conclusion section.
 18. Captions of the figures must be with complete information, experimental conditions etc
 19. Make sure all analysis was done at least three times with no significant difference
 20. Quality of figures is to be improved, please improve the figures
 21. Values in the tables should be of uniform significant figures, please recheck
- I WOULD LIKE TO SEE THE REVISED MANUSCRIPT.

Reviewer: 3

Comments to the Author(s)

In this manuscript, the authors presented work on designing and synthesizing the copolymer containing salicylic acid (SA) motifs via polycondensation method. It seems that they successfully obtained the copolymer and evaluated their basic properties. The data presented in the paper are of relatively high quality. After evaluating the overall topic of the work, which seems to address an important of the field, however, several issues need to be clarified before accepted in this journal.

- a) Please show the GPC traces in the manuscript.
- b) Please polish your writing, for example, the sentence "The degradation rate of P(BS-co-SA) was high in enzymatic solution" in the manuscript should be deleted.
- c) The author stated that all samples were replicated thrice, please add error bar in the Tables and Figure 5.
- d) The author mentioned "spiral chain structure of PBS...", please add relative references.
- e) To intuitively grasp the properties difference of different samples, the author should used the experimental data to state the properties change, for example, "The tensile strength of the copolymers was less than pure PBS except for P(BS-co-0.5%SA)"; "Figure 3A and 3B exhibit the DSC curves measured with pure PBS and P(BS-co-SA). P(BS-co-SA) copolymers were observed to have similar T_m and a high T_c compared with pure PBS," there are too many sentences should be stated using the experimental data.
- f) please explain this sentence "but their location is offset in the direction of the large angle. It suggested that their crystal structures are similar. However, crystal behavior is not similar".

Author's Response to Decision Letter for (RSOS-190504.R0)

See Appendix A.

RSOS-190504.R1 (Revision)

Review form: Reviewer 2

Is the manuscript scientifically sound in its present form?

Yes

Are the interpretations and conclusions justified by the results?

Yes

Is the language acceptable?

Yes

Is it clear how to access all supporting data?

Yes

Do you have any ethical concerns with this paper?

No

Have you any concerns about statistical analyses in this paper?

No

Recommendation?

Accept as is

Comments to the Author(s)

The authors improved the paper

Decision letter (RSOS-190504.R1)

11-Jun-2019

Dear Professor Zhang:

Title: Poly(butylene succinate-co-salicylic acid) Copolymers and their Effect on Promoting Plant Growth

Manuscript ID: RSOS-190504.R1

It is a pleasure to accept your manuscript in its current form for publication in Royal Society Open Science. The chemistry content of Royal Society Open Science is published in collaboration with the Royal Society of Chemistry.

RSC Associate Editor:
Comments to the Author:
(There are no comments.)

RSC Subject Editor:
Comments to the Author:
(There are no comments.)

Reviewer(s)' Comments to Author:
Reviewer: 2

Comments to the Author(s)
The authors improved the paper

Appendix A

Dear Editor, Dear reviewers

Thank you for your letter dated 30 April. We were pleased to know that our work was rated as potentially acceptable for publication in Journal, subject to adequate revision. We thank the reviewers for the time and effort that they have put into reviewing the previous version of the manuscript. Their suggestions have enabled us to improve our work. Based on the instructions provided in your letter, we uploaded the file of the revised manuscript. Accordingly, we have uploaded a copy of the original manuscript with all the changes highlighted.

Appended to this letter is our point-by-point response to the comments raised by the reviewers. The comments are reproduced and our responses are given directly afterward in a different color (red).

We would like also to thank you for allowing us to resubmit a revised copy of the manuscript.

We hope that the revised manuscript is accepted for publication in Royal Society Open Science.

Sincerely,

Corresponding Author:

Prof. Min Zhang

Shaanxi Key Laboratory of Chemical Additives for Industry,

Shaanxi University of Science and Technology,

Xi'an 710021, China

E-mail: yanjiushi206@163.com

Reviewer: 1

1. The authors should clarify how to remove $Ti(OBu)_4$ when purified P(BS-co-SA) product.

Thank you for underlining this point. We did not remove $Ti(OBu)_4$ when purified P(BS-co-SA) product. Currently, $Ti(OBu)_4$ is being used as a catalyst in PBS synthesis, it cannot be removed. Therefore, we use other methods, such as organic catalyst instead of inorganic catalyst, and keep the dose as low as possible to reduce pollution.

2. The authors should give the detailed information for the BSB, SaBS, SaSB...

Thank you for the suggestion. We have detailed the information according to the comment. (page 4)

BSB, SaBS, SaSB, SaSSa and BSaS represent different molecular fragments in the polymer chain, respectively. Where B is 1, 4-butanediol, S is succinic acid, and Sa is salicylic acid.

3. The authors should give the degradation mechanism of PBS and P(BS-co-SA) for readers understanding.

Thank you for the suggestion. We have added the reference of degradation mechanism to article according to the suggestion. (page 7)

45. Zhang M, Ma X, Li C, Zhao D, Xing Y, Qiu J. 2017 A correlation between the degradability of

poly(butylene succinate)-based copolyesters and catalytic behavior with *Candida antarctica* lipase
B. *RSC Adv.* 7, 43052-43063. (doi: 10.1039/c7ra05553f)

Reviewer: 2

1. The title should be improved; it should be more precise and represent to the contents.

The title has been revised to Poly(butylene succinate-*co*-salicylic acid) Copolymers and their Effect on Promoting Plant Growth.

2. In the abstract; “proved that the degradability of copolymers was more.” add quantitative data.

Thank you for the suggestion. We have rewrote the sentence according to the comment. (page 1)

By molecular docking simulations, it was proved that the degradability of copolymers was more effective than that of pure PBS with binding energy of $-5.77 \text{ kcal}\cdot\text{mol}^{-1}$.

3. The abstract should be re-written to summarize the work; the abstract should state briefly the purpose of the research, the principal results and major conclusions. An abstract is often presented separately from the article, so it must be able to stand alone.

Thank you for the suggestion. We have rewrote the abstract according to the suggestion. (page 1)

Biodegradable random copolymers was successfully synthesized by melt polycondensation of poly(butylene succinate) (PBS) and salicylic acid (SA). The obtained copolymers was characterized by proton nuclear magnetic resonance

spectroscopy. The effect of different SA contents on the properties of copolymers was investigated by universal testing machine, thermogravimetric analyzer, differential scanning calorimetry and X-ray diffraction analysis. The results showed that the copolymers with 0.5% SA contents exhibit excellent elastic modulus (1413.0 MPa) and tensile strength (192.8 MPa), and the similar thermal decomposition temperature (≈ 320 °C) compared with pure PBS. By molecular docking simulations, it was proved that the degradability of copolymers was more effective than that of pure PBS with binding energy of -5.77 kcal \cdot mol $^{-1}$. PBS copolymers with a small amount of SA were not only biodegradable but could stimulate the growth of green vegetables. So, biodegradable polymers can be use in wide range as they are environmental friendly.

4. The Introduction should be focused and goes over the topic.

Thank you for the suggestion. We have revised the introduction according to the suggestion. (page 2)

5. Introduction should increase scientific understanding sufficiently, to be of a very high quality paper, what is the gap to cover.

Thank you for the suggestion. We have revised the introduction. (page 2)

For decades, traditional polymeric materials have been widely used in many fields. Among them, packaging has a considerable market segment. As a result, millions of tons of plastic packaging are landfilled every year [1] and cause environmental pollution. This led to the research that focused on developing biodegradable materials [2,3]. Other environmental friendly substances are now gradually replacing many harmful chemical compounds, like petroleum hydrocarbons [4-7].

6. All abbreviations should be defined.

The new abbreviations are Salicylic acid (SA), Proton nuclear magnetic resonance

spectroscopy ($^1\text{H NMR}$) and Wide-angle X-ray diffraction (XRD). (page 3)

In agriculture, SA acts as an endogenous signal molecule in plants growth. (page 2)

7. Page 5 Section 2.2. Synthesis of copolymers, should be revised with more details and steps are to be clear.

Thank you for the suggestion. We have revised the information with more details and steps. (page 3)

PBS-based copolymers were prepared by the melt polycondensation method as reported earlier [32], including two steps of esterification and polycondensation. Succinic acid, 1,4-butanediol and SA as raw material was added into the three-necked flask at 1:1.1 (acid to alcohol), then 0.5 mol% of $\text{Ti}(\text{OBU})_4$ was added as the catalyst for the reaction. The reactor was filled with nitrogen to protect the reaction from oxidation. The reactants were heated to melt in the oil bath at 190 °C, until the volume of dehydrate and the theoretical amount of H_2O become equal. The phenyl functional groups from SA were added to the polymer chain during dehydration reaction. Then the polycondensation process was carried out controlling the pressure to below 5.5 kPa (vacuum degree). It was continued to heat up until 220°C and reacted for approximately 3 h to obtain desired products. Products were dissolved in CHCl_3 , and purified several times with ethanol, and then vacuum dried for 24 h.

8. Scheme 1 is to be revised, add all conditions, T, used gas, media, aqueous / not, time, initiator, etc.

Scheme 1. Synthesis route of the copolymers: P(BS-*co*-SA)

9. Section 2.3. Characterizations. Please add more details about the procedure; check the numbers. Be more specific, what about the control set.

Chloroform was used as the mobile phase at a 1.0 mL min⁻¹ flow rate and the sample solution was 1.0 w/v%. The column temperature was maintained at 40 °C in all analyses. (page 3)

About 3 mg of sample was heated from 25 °C to 200 °C at a rate of 30 °C min⁻¹ in order to eliminate any previous thermal history. (page 4)

Polarized optical microscope (POM) was used with a DK-POL (Chongqing Optec, China) in order to observe the crystalline morphologies of the copolymers. Samples by dipping polymers in chloroform solution on the glass slide and the sample film was obtained after evaporation of the solvent. The samples were heated until melt and then temperature was maintained to 80 °C for 3 h. (page 4)

10. Please correct the errors and typos.

Capacity of copolymers to improve plant growth was also investigated. (page 2)

We have corrected the “decom position” to “decomposition”. (page 6)

^b The crystallinity (χ_c) of copolymers were calculated by $\chi_c = \Delta H_m / \Delta H_m^o \times 100\%$, and ΔH_m^o is the melting enthalpy of 100% crystalline PBS which is assumed to be 110.5 Jg⁻¹. (page 16)

11. Please correct the equations.

Thank you for the suggestion. We have correct the equations required. (page 4)

$$\% \text{ weight loss} = \frac{W_o - W_t}{W_o} \times 100\%$$

12. Please improve Fig. 2. with labeling and descriptions; please revise the interpretation and correct.

We have improved Figure 2 according to the suggestion, and the interpretation and correct have also been revised.

Figure 2. (A) ¹H NMR spectrum of the copolymers: P(BS-co-SA)

13. Please improve the discussion of molecular docking.

Thank you for the suggestion. We have improved the discussion according to the suggestion. (page 7)

The interactions between CALB lipase and ligands were explained at the molecular level by molecular docking simulations. The binding free energy was calculated and the relationship among all related energies is clearly shown in Table 4. Among them the $E_{\text{torsional}}$ only depends on the number of torsion bond and E_{unbond} is equal to

$E_{\text{total-internal}}$. In accordance with the molecular docking results, the binding free energy of compounds formed by substrates (SaBS, SaSB, SaSSa and BSaS) with CALB lipase were -5.00, -5.48, -5.77 and -4.40 kcal·mol⁻¹ respectively, which were lower than that of BSB, -4.04 kcal mol⁻¹. The lower the binding energy, the more stable the substrate binding to the enzyme [45]. The binding free energy of SaSSa with CALB lipase was the lowest, so SaSSa was the most stable with CALB lipase. Then the substrate containing SaBS, SaSB and BSaS unit was relatively stable with CALB lipase owing to the lower binding energy. It can be concluded from the above results that the presence of Sa unit could significantly reduce binding energy and enhanced the degradability of PBS in phosphate buffer solution. Therefore, the enzymatic degradability of P(BS-*co*-SA) was more effective than that of PBS.

14. Effects on Vegetable Growth: please discuss it deeply as it is important section.

Thank you for the suggestion. We have improved the discussion according to the suggestion. (page 8)

We investigated the effect of soil-incorporated copolymers' residues on growth of plants. The effect of different levels of soil incorporated P(BS-*co*-SA) copolymer on vegetable growth was studied (Figure 6). Three levels (0g, 0.1g and 0.6g) of P(BS-*co*-SA) were used during the study. Soil incorporated without P(BS-*co*-SA) was used as control soil. It can be seen from Figure 6 that no significant change in plant growth was observed during the first 20 d. At day 30, number of leaves recorded in A-2 were higher than in B-2 and C-2. But with the time, treatments containing P(BS-*co*-SA) showed higher growth rate as compared to the treatments without P(BS-*co*-SA) (control treatment). This was because P(BS-*co*-SA) undergo little or no degradation in the early days because release of SA into the soil from the polymer took time [36]. At day 60, it was found that the number of leaves in all treatments were same but the plants on the soil treated with copolymers exhibited larger and green leaves as compared to control. This became more obvious with the increase in

copolymer content. After a period of time, the P(BS-co-SA) hydrolyzes upon exposure to aqueous conditions in soil, releasing SA while leaving the PBS backbone fracture for degradation [47]. The vegetable growth for different SA content showed that the polymer with the highest content of SA had a better relative growth rate. P(BS-co-SA) stimulated plant growth and no toxicity was observed. It is well-known that SA could induce vegetable growth. The results reported here agree with this consensus.

15. There are many sentence that required to be clarified.

Thank you for the suggestion. We have revised the sentences.

The intensities of P(BS-co-SA) diffraction peaks are the same as that of pure PBS, meanwhile, the locations of diffraction peaks are almost the same. It suggested that their crystal structures are similar. (page 6)

The degradability of P(BS-co-SA) films in basic media was studied as shown in Figure 5. It was found in enzymatic degradation that % weight loss of P(BS-co-SA) was significantly higher as compared to pure PBS. At day 20, % weight loss of P(BS-co-5SA) was recorded as 3.38% which is considerably higher than 1.53% of pure PBS. It was because the spiral chain structure of PBS had been destroyed and the crystallinity of PBS was changed in the presence of SA. (page 7)

16. Under introduction after “Many environmentally unfriendly chemical compounds like petroleum hydrocarbons” please add [1] Ecological Engineering 120, 126-133; [2] IGI ISBN13: 978152252136; [3] Elsevier ISBN-13: 978-0128047033; [4] Journal of Environmental Chemical Engineering 6 (4), 5361-5368; [5] Journal of environmental management 211, 323-333.

Thank you for the suggestion. We have added references required in introduction.

Other environmental friendly substances are now gradually replacing many harmful chemical compounds, like petroleum hydrocarbons [4-7]. (page 2)

17. More emphasis on finding and its implication may be mentioned in the conclusion section.

Thank you for the suggestion. We have rewrote the conclusion according to the suggestion. (page 8)

A series of P(BS-*co*-SA) copolymers were synthesized successfully by a two-step melt polycondensation reaction, which has a superior ability to induce plant growth compared to PBS. The effects of low, medium and high SA content on copolymers performance were investigated. Tensile tests and thermal evaluation showed that copolymers had good mechanical and thermal properties when a small amount of SA was added. Thermal stability up to 300 °C was confirmed by TGA. XRD indicated that the copolymers with different SA contents had similar crystal structure. The enzymatic degradation of P(BS-*co*-SA) was faster than that of PBS in phosphate buffer solution. The enzymatic degradation fit well with the molecular docking results, the presence of SA unit could significantly enhance the degradability of PBS due to the lower free binding energy, indicating the favorability of adding SA. These results suggested that adding a small amount of SA would not affect the comprehensive performance of PBS, meanwhile, it could stimulate the growth of green vegetables and can be used as environmental friendly material.

18. Captions of the figures must be with complete information, experimental conditions etc.

Thank you for the suggestion, we have modified captions of the figures throughout the text.

Table 1. Composition and relative parameters^a of the copolymers: P(BS-*co*-SA).

^a Obtained by GPC calibrated by polystyrene standards. ^b Number average molecular weight. ^c Weight average molecular weight. ^d Polydispersity index.

Table 3. Thermal performance of various P(BS-co-SA) determined by TGA and DSC.

^a Determined by the second heating scan at 10 °C min⁻¹. ^b The crystallinity (χ_c) of copolymers were calculated by $\chi_c = \Delta H_m / \Delta H_m^o \times 100\%$, and ΔH_m^o is the melting enthalpy of 100% crystalline PBS which is assumed to be 110.5 Jg⁻¹ [39]. ^c Decomposition temperature of copolymers at weight loss of 5%. ^d Decomposition temperature of copolymers at weight loss of 50%.

Table 4. Docking results of substrate molecules by AutoDock program.

The binding results were obtained by analyzing the 200 docking poses performed by AutoDock 4.2. ^a Binding energy. ^b Intermolecular energy. ^c Total internal energy. ^d Torsional energy. ^e Van der Waals energies. ^f Electrostatic energy.

19. Make sure all analysis was done at least three times with no significant difference.

Thank you for underlining these analysis, and we make sure it was done at least three times with no significant difference. We have added error bar in Table 2 and Figure 5.

20. Quality of figures is to be improved, please improve the figures.

We are grateful for the suggestion, and we have improved the quality of figures.

21. Values in the tables should be of uniform significant figures, please recheck.

Thank you for the suggestion, we have rechecked values throughout the tables.

Table 2. Mechanical properties of the P(BS-co-SA) films.

polymers	Elastic modulus [MPa]	Tensile strength [MPa]	Elongation at break [%]
PBS	779.6 ± 4.65	180.5 ± 2.87	13.2 ± 1.54
P(BS-co-0.5%SA)	1413.0 ± 5.04	192.8 ± 2.43	11.3 ± 1.98
P(BS-co-5%SA)	1434.7 ± 4.18	87.9 ± 1.18	2.7 ± 2.12
P(BS-co-10%SA)	----	----	----

('-' indicates film that was too fragile to be tested)

Table 3. Thermal performance of various P(BS-co-SA) determined by TGA and DSC.

polymer	T_g (°C)	T_m (°C)	ΔH_m^a (Jg ⁻¹)	χ_c^b (%)	T_c (°C)	ΔT (°C)	$T_{5\%}^c$ (°C)	$T_{50\%}^d$ (°C)
PBS	-28.5	116.4	76.6	69.3	79.4	37.0	321.2	379.0
P(BS-co-0.5%SA)	-28.2	116.7	72.9	66.0	83.2	33.5	320.6	377.4
P(BS-co-5%SA)	-26.9	116.5	75.0	67.9	80.3	36.2	304.3	362.8
P(BS-co-10%SA)	-29.0	115.2	75.7	68.5	77.0	38.2	303.2	357.3

Reviewer: 3

a) Please show the GPC traces in the manuscript.

We have added the GPC traces in the manuscript. (Figure 2B)

Figure 2. (B) GPC diagrams of the copolymers: P(BS-co-SA)

b) Please polish your writing, for example, the sentence “The degradation rate of P(BS-co-SA) was high in enzymatic solution” in the manuscript should be deleted.

Thank you for the suggestion. We have revised the writing and the sentence has been deleted according to the comment. (page 7)

c) The author stated that all samples were replicated thrice, please add error bar in the Tables and Figure 5.

Figure 5. The degradation evaluation of P(BS-co-SA) in phosphate buffer solution containing CALB

Table 2. Mechanical properties of the P(BS-co-SA) films.

polymers	Elastic modulus [MPa]	Tensile strength [MPa]	Elongation at break [%]
PBS	779.6 ± 4.65	180.5 ± 2.87	13.2 ± 1.54
P(BS-co-0.5%SA)	1413.0 ± 5.04	192.8 ± 2.43	11.3 ± 1.98
P(BS-co-5%SA)	1434.7 ± 4.18	87.9 ± 1.18	2.7 ± 2.12
P(BS-co-10%SA)	---	---	---

d) The author mentioned “spiral chain structure of PBS...”, please add relative references.

Thank you for the suggestion. We have added the reference to article according to the suggestion. (page 7)

44. Morris GM, Huey R, Lindstrom W, Sanner MF, Belew RK, Goodsell DS, Olson AJ. 2009 AutoDock4 and AutoDockTools4: Automated Docking with Selective Receptor Flexibility. *J. Comput. Chem.*, **30**, 2785-2791. (doi: 10.1002/jcc.21256)

d) To intuitively grasp the properties difference of different samples, the author should use the experimental data to state the properties change, for example, "The tensile strength of the copolymers was less than pure PBS except for P(BS-co-0.5%SA)"; "Figure 3A and 3B exhibit the DSC curves measured with pure PBS and P(BS-co-SA). P(BS-co-SA) copolymers were observed to have similar T_m and a high T_c compared with pure PBS," there are too many sentences should be stated using the experimental data.

1. An apparent increase in elastic modulus of the P(BS-co-SA) films from 779.6 MPa to 1400 MPa was observed when SA was added, but there was significant reduction in elongation at break from 13.2% to 2.7%. The tensile strength of the P(BS-co-0.5%SA), 192.8 MPa, was more than that of pure PBS (180.5 MPa).

(in 3.2. Mechanical properties) (page 5)

2. It is observed that pure PBS displays a crystallization peak (T_c) at 79.4 °C, an endothermic peak (T_m) at 116.4 °C and a glass transition temperature (T_g) at -28.5 °C. P(BS-co-0.5% SA) was observed to have a high T_c of 83.2 °C compared with pure PBS. The T_g (-28.2 °C) and T_m (116.7 °C) of P(BS-co-0.5%SA) were found similar to pure PBS, but ΔT (33.5 °C) was lower than that of pure PBS. These results suggested that the thermal stability of PBS was significantly enhanced when 0.5% SA was added. However, T_m (115.2 °C), T_c (77.0 °C) and T_g (-29.0 °C) of P(BS-co-10%SA) were less as compared to PBS, which may be attributed to its lower molar mass.

(in 3.3. Thermal analysis) (page 6)

3. The degradability of P(BS-co-SA) films in basic media was studied as shown in Figure 5. It was found in enzymatic degradation that % weight loss of P(BS-co-SA) was significantly higher as compared to pure PBS. At day 20, % weight loss of

P(BS-*co*-5SA) was recorded as 3.38% which is considerably higher than 1.53% of pure PBS. It was because the spiral chain structure of PBS had been destroyed and the crystallinity of PBS was changed in the presence of SA. (in 3.5. Enzymatic degradation) (page 7)

e) please explain this sentence “but their location is offset in the direction of the large angle. It suggested that their crystal structures are similar. However, crystal behavior is not similar”.

Thank you for underlining this confusion. Maybe the sentence was confusing, so I rewrote it. (page 6)

The intensities of P(BS-*co*-SA) diffraction peaks are the same as that of pure PBS, meanwhile, the locations of diffraction peaks are almost the same. It suggested that their crystal structures are similar.